# Co-Culture of Mesenchymal Stem Cells and Ligamentocytes on Triphasic Embroidered Poly(L-lactide-co-ε-caprolactone) and Polylactic Acid Scaffolds for Anterior Cruciate Ligament Enthesis Tissue Engineering

**DOI:** 10.3390/ijms24076714

**Published:** 2023-04-04

**Authors:** Clemens Gögele, Julia Vogt, Judith Hahn, Annette Breier, Ricardo Bernhardt, Michael Meyer, Michaela Schröpfer, Kerstin Schäfer-Eckart, Gundula Schulze-Tanzil

**Affiliations:** 1Institute of Anatomy and Cell Biology, Paracelsus Medical University, Nuremberg and Salzburg, Prof. Ernst Nathan Str. 1, 90419 Nuremberg, Germany; 2Department Materials Engineering, Institute of Polymers Materials, Leibniz-Institut für Polymerforschung Dresden e.V. (IPF), Hohe Straße 6, 01069 Dresden, Germany; 3FILK Freiberg Institute gGmbH (FILK), Meißner Ring 1-5, 09599 Freiberg, Germany; 4Bone Marrow Transplantation Unit, Medizinische Klinik 5, Klinikum Nürnberg, 90419 Nuremberg, Germany

**Keywords:** enthesis, tissue engineering, anterior cruciate ligament, co-culture, medical lace embroidery, poly(L-lactide-co-ε-caprolactone) (P(LA-CL))/polylactic acid (PLA), triphasic scaffold

## Abstract

Successful anterior cruciate ligament (ACL) reconstructions strive for a firm bone-ligament integration. With the aim to establish an enthesis-like construct, embroidered functionalized scaffolds were colonized with spheroids of osteogenically differentiated human mesenchymal stem cells (hMSCs) and lapine (l) ACL fibroblasts in this study. These triphasic poly(L-lactide-co-ε-caprolactone) and polylactic acid (P(LA-CL)/PLA) scaffolds with a bone-, a fibrocartilage transition- and a ligament zone were colonized with spheroids directly after assembly (DC) or with 14-day pre-cultured lACL fibroblast and 14-day osteogenically differentiated hMSCs spheroids (=longer pre-cultivation, LC). The scaffolds with co-cultures were cultured for 14 days. Cell vitality, DNA and sulfated glycosaminoglycan (sGAG) contents were determined. The relative gene expressions of *collagen types I* and *X*, *Mohawk*, *Tenascin C* and *runt-related protein (RUNX) 2* were analyzed. Compared to the lACL spheroids, those with hMSCs adhered more rapidly. Vimentin and collagen type I immunoreactivity were mainly detected in the hMSCs colonizing the bone zone. The DNA content was higher in the DC than in LC whereas the sGAG content was higher in LC. The gene expression of ECM components and transcription factors depended on cell type and pre-culturing condition. Zonal colonization of triphasic scaffolds using spheroids is possible, offering a novel approach for enthesis tissue engineering.

## 1. Introduction

Anterior cruciate ligament ruptures are still among the most common ligament injuries in the knee [1,2]. Ruptures occur preferentially near the osteochondral insertion site or enthesis [3]. The femoral and tibial enthesis of the anterior cruciate ligament (ACL) are transition zones and connect the bones to the ligamentous midsubstance tissue via fibrocartilage [4,5,6]. Histologically, a typical fibrocartilaginous tendon or ligament enthesis comprises a change in cell shape (spindle-shaped ligamentocytes in the midsubstance convert into ovoids/round fibro-chondrocytes as well as become osteoblasts in the bone insertion), the presence of a tidemark [7] followed by a mineralization gradient and changed orientation of collagen fibers [8]. Mechanical cues and certain biological factors are essential for the formation of a functional and thus mechanically resilient graded ECM in the enthesis due to the different expression of ECM components [9]. The self-healing capacity of the enthesis after injuries is reduced, and one hypothesis is that this could be the result of low cell density in this area [10,11,12]. Another reason for this result is the low vascularization of this tissue, with avascularity in the fibrocartilaginous zones [8]. 

ACL reconstructions should be performed to avoid further damages/abnormities of the surrounding tissues such as the articular cartilage leading to osteoarthritis [13,14], menisci roots [15], subchondral bone [16], infrapatellar fat pad, changes of the synovial fluid [17] or the collateral ligaments [18] resulting in their degeneration. The current gold standard is to substitute the ruptured ACL with ligament autografts most often harvested from the so-called Hamstring, or from the quadriceps muscles or using the patellar ligament [19]. However, autografts from the Hamstring muscles are limited and, due to the fact that a higher incidence of tunnel widening and electromechanical delay in the knee flexors happen after autologous reconstructions [13,20], there is an urgent need for tissue-engineered ligament grafts with a special attention to the different zoning at the future bone-transition area [21]. This is because the current gold standard (Hamstring) uses only tendinous tissue, but it has been shown that its bony integration remains insufficient [22]. Many artificial ACL products (Leeds-Keio^TM^, the LARS^TM^ [Ligament Advanced Reinforcement System] and the Trevira Hochfest^TM^) have passed the pre-clinical and clinical phases and are now on the market, but there is still discontent among surgeons concerning their suitability [20,23]. Although many of them are biocompatible and allow a cell colonization with ECM formation, the outcome is in some cases still not sufficient [24]. One problem with synthetic products is that they do not completely reflect the complex three-dimensional (3D) structure of the enthesis associated with a specific mechanical behavior [8,25,26,27]. For this reason, an attempt in our study was undertaken to make the geometry of the scaffold very similar to the native ACL. The influence of the scaffold topology (porosity, thickness, filament dimensions, structure and surface modification) is not only crucial for cellular differentiation and its ECM deposition, but also for withstanding mechanical stress [28,29]. Using embroidery technology, especially medical lace embroidery, as a scaffold fabrication technique seemed to be a promising methodology in this reconstructive tissue engineering field to achieve a mechanically resilient and zonal tissue [30]. Compared to other methods such as 3D bioprinting [31,32], electrospinning [33,34,35] or braiding [36], medical lace embroidery works with the interlacing of highly oriented thread materials and therefore enables the fabrication of stress-adapted structures [37]. The synthetic materials poly(l-lactide-co-ε-caprolactone) (P(LA-CL)) and polylactic acid (PLA) used in the present study are well established in the medical field [38,39] and also proven to be cyto- and biocompatible after functionalization [40,41,42]. Functionalization was reached with a two-step method; the first comprised a gas-phase fluorination for reaching a high cellular adherence on the scaffolds [43] followed by an additional collagen foam infiltration step [44]. In order to achieve a targeted seeding with a direct colonization of the different zones, self-assembled spheroids were chosen for seeding. In the present study, we investigated the precise positioning of lapine anterior cruciate ligamentocyte (lACL) and human mesenchymal stem cell (hMSC) spheroids on functionalized embroidered triphasic scaffolds and therefore, the simultaneous application of different cell types on the same material, the scaffold vitalization and expression of zone-specific ECM formations. It has to be mentioned that the focus of this study was still on the bone and the ligament zones and not on the fibrocartilage zone. The aim of this study was initially to develop a simple co-culture model with only two zones in vitro since in vivo, it is believed that fibrocartilage forms automatically in the transition zone of scaffold, probably due to the natural stresses under mechanical loading [45,46,47,48]. Thus, for the time being, the bone zone was the main focus of interest as it is absolutely essential for the mechanical/loosening-free anchorage of the implant. In particular, the use of stem cells has become an indispensable part of regenerative therapy due to their promising and far-reaching capabilities. Nevertheless, there are still many unanswered questions regarding the mutual influence of different cell types and how this affects ECM formation. 

Therefore, our hypothesis is that a longer osteogenic pre-differentiation of hMSC spheroids is more effective in regard to osteogenic commitment of hMSCs than a direct colonization strategy. For ACL tissue engineering, the question now is whether a triphasic three-dimensional (3D) scaffold structure can be vitalized quickly and successfully using a directed seeding strategy with spheroids.

## 2. Results

### 2.1. Porosity of Triphasic Embroidered Scaffolds

The triphasic scaffolds reflected the dimensions of a rabbit ACL [49] with a length of 18 mm, width of 4 mm and thickness of 2 mm (Figure 1A represents an ACL half consisting of a bony (a), plus a transition (b) and half a ligament (c) area). 

The total area of the triphasic scaffold was 72 × 106 µm^2^. The bone zone possessed dimensions of 22.4 × 106 µm^2^, and the ligament zone represented an area of 28.6 × 106 µm^2^. The investigation of the morphological properties by means of computed tomography revealed for the triphasic embroidered scaffolds, a porosity for the bone zone of 86.8 ± 2.0% (Figure 1(Ba)), for the transition 62.6 ± 1.0% (Figure 1(Bb) and for the ligament zone, 68.4 ± 0.9% (Figure 1(Bc)). The mean pore equivalent diameter was, for the bone region, 298.8 ± 146.9 µm, for the transition region, 151.3 ± 75.2 µm and for the ligament region, 155.7 ± 88.4 µm (Figure 1C,D).

### 2.2. Spheroid Formation, Vitality and Changes in Size and Shape

For a successful colonization of embroidered triphasic scaffolds, the used cell spheroids should be vital. Therefore, a vitality assay was performed to distinguish between vital and dead cells of the spheroids before seeding them on the scaffold zones. Only a few dead cells could be detected directly after spheroid formation not only in the hMSC spheroids but also in the lACL spheroids (Figure 2A,C). After a further 14 days of cultivation (LC), more red dots could be detected, and a change in spheroid size could be recognized in the hMSC spheroids, but also in the lACL spheroids (Figure 2B,D). No significant differences between the two different cell types and the cultivation time could be calculated either in the vitality, in the diameter or in the roundness. In general, the cell vitality of the spheroids was higher than 90%, but the vitality in those consisting of lACLs was higher in comparison to the hMSC spheroids not only in the DC but also in the LC (not significant, Figure 2E). The diameter of the hMSC spheroids directly after formation and before further cultivation was 612 ± 96 µm, and the lACL spheroids had a diameter with a mean of 613 ± 175 µm. During the longtime cultivation with osteogenic medium, the hMSC spheroids enlarged slightly to 627 ± 183 µm, while the lACL in their normal growth medium became smaller (422 ± 85 µm) (not significant, Figure 2F). Nevertheless, all spheroids retained their round shapes. This was particularly evident in the measurement of roundness; no significant differences were found between cell types or between cultivation durations. The roundness was above 0.8 for all of them (Figure 2G). 

### 2.3. Vitality of Co-Colonized Scaffolds

Cytocompatibility of triphasic scaffolds is a requirement for successful tissue engineering. For this reason, a viability assay after 14 days of static co-cultivation was performed. More green cells could be observed in the direct culture on the bone zone with the hMSC spheroids in comparison to the ligament zone with the lACL spheroids. The accumulation of dead cells in the lower third (marked with red stars) of the picture were due to the bisection process by scalpel (Figure 3A). The 14 day-osteogenically differentiated hMSC spheroids also adhered on the bone zone of the scaffolds and were still alive after a further 14 days of cultivation. Only a few vital lACLs could be observed in the long-time cultivation (Figure 3B). Based on the vitality pictures, the colonized surface could be calculated and showed that the bone zone seeded directly with hMSC spheroids represented the largest area covered with cells. The long-time cultivation of both cell types led to a reduction of the colonized zones (not significant, Figure 3C). A higher magnification of the images shows much clearer that the single cells (spindle-shaped hMSCs) and lACL fibroblasts migrated out of the spheroid cell cluster and colonized the single polymer threads. Cell vitality of the hMSCs and lACL fibroblasts was higher than 60% in the DC and LC (Appendix A). 

### 2.4. Expression of Collagen Type I and Vimentin of hMSCs and lACLs in Bone and Ligament Scaffold Zones and Cell Proliferation

Collagen type I and vimentin could be detected not only in the bone but also in the ligament zone colonized with directly formed spheroids (DC) or longer pre-cultured spheroids (LC). Both proteins were mostly homogenously distributed around the spheroids and their emigrating cells. More immunoreactivity could be detected in the DC-colonized scaffold areas not only in the bone but also in the ligament zone (Figure 4). It has to be mentioned also that the collagen foam, which is not always homogenously distributed on the scaffold, showed some immunoreactivity for collagen type I. Images at higher magnification allow a detailed view of the spheroids (Sph) and the single cells on the polymer threads and prove that both cell types express collagen type I and vimentin in the DC and the LC (Appendix A). The scaffolds were stained for F-actin and immunolabeled for the proliferation marker Ki-67. The spheroid integration on the scaffold surface and the strong filamentous actin network of both cell types on the scaffold threads could be seen. Furthermore, both cell types (especially the cells at the border of the spheroids) are proliferative active in the DC and the LC (Appendix A). 

### 2.5. DNA and Sulphated Glycosaminoglycan Contents in Bone and Ligament Scaffold Zones 

To estimate whether there was a cell proliferation on the scaffolds during long-time cultivation, the DNA amount was quantitatively measured. In general, the DNA was higher in the bone zone colonized with hMSC spheroids in comparison to the ligament zone colonized with lACL spheroids. The DNA amount in the triphasic scaffolds co-cultured with hMSC and lACL spheroids 2 days after formation had a higher DNA content in contrast to the longer pre-cultivated spheroids (Figure 5A). Neither in the DNA nor in the sGAG content could significant differences be calculated, but a longer cultivation led to increased mean values of the sGAG content, especially regarding the lACL spheroids, which had the highest sGAG amount after 14 days of spheroid formation plus 14 days on the ligament zone (Figure 5B). 

### 2.6. Relative Gene Expression of Zonal Specific Extracellular Matrix Components and Transcription Factors

The gene expressions of main bone and ligament ECM components such as *collagen type I*, *type X* and *tenascin C* and the bone- and ligament-related transcription factors, *runt-related protein (RUNX)2* and *Mohawk (MKX)* were evaluated. The hMSC spheroids required a longer pre-cultivation to express more *collagen type I*, while no differences could be observed in the *RUNX2* expression. The relative gene expression of *collagen type X* was higher in direct culture of hMSC spheroids in the bone zone in comparison to the long-time pre-cultivation. The relative gene expression of the ligament-related ECM components, *collagen type I* and *tenascin C* was higher in the direct culture in comparison to the long-time pre-cultivation. The relative gene expression of the ligamentous transcription factor *MKX* was very low after both cultivation strategies in comparison to collagen type I and tenascin C in the ligament zone (Figure 6).

### 2.7. Progression of Calcium Deposition 

The scaffolds co-cultured with directly formed or long-time pre-cultivated spheroids were stained as a whole with Alizarin Red to observe calcium depositions and therefore the beginning of an ossification in the bone zone. Light microscopical images showed that hMSC and lACL spheroids were attached to the scaffold in response to the direct cultivation strategy and also the emigrating cells (Figure 7A–H). The amount and the size of long-time pre-cultivated spheroids were reduced but the mean intensity of red was higher in comparison to the directly colonized spheroids, and the osteogenically differentiated hMSCs showed the highest red intensity (Figure 7I).

## 3. Discussion

Most ACL injuries occur in both non-athletic and athletic individuals between the ages of 20 and 24 [2]. For this reason, reconstructions are of crucial importance, not only to restore sufficient knee stability, but above all to be able to be active in sports again. Although the autologous graft materials used today are successful, the re-rupture rate is still too high (overall 4.8% depending on the surgery technique) [50], and therefore, the orthopedic research community is striving to find a tissue-engineered construct as an option to overcome the problem of limited autograft availability. In the past few years, new approaches and techniques, such as modification of decellularized native enthesis tissue, silk [51,52,53] or polymer based scaffolds [25,54], have been investigated for their reconstruction capabilities for the complete enthesis zone and not only the ligament zone of the ACL, the patellar tendon or the rotator cuff [25,51,52,53]. The importance of creating multiphasic scaffolds for musculoskeletal interface tissue engineering to achieve functional restoration of injured zonal tissues has increased in past years not only in orthopedics but also in dentistry [55,56,57,58]. Another method involves the use of embroidery technology that enables the fabrication of complex structures whose morphological and mechanical properties can be specifically adjusted according to pattern designs. In addition, the use of different polymer thread materials and gas-phase fluorination, which could enhance cell adhesion as shown for fibroblasts [43], is a promising approach for ACL reconstruction [39]. Not only the cytocompatibility [44,59], but also the biocompatibility of functionalized embroidered scaffolds have been proven in recent years [42]. Further development of the embroidered scaffolds mentioned above, consisting only of the pattern design for the ligament, was performed in the present study creating topologically adapted cartilage and bone zones, which should result in an improved adaptation to the native anatomical conditions of the enthesis in the knee joint. Pitta-Kruize et al. described the bone–ligament interface as multi-material interface that carries loads using structural mechanisms such as the overlap of collagen fibers intruding into the bone. This achieves a firm hold between the stiff bone and the more elastic ligament while allowing for an effective transition between these materials [60,61]. The gradual design with undulating interfaces to the transition area of the triphasic embroidered scaffolds addresses this structural mechanism in the present study. It should be noted that the determination of pore sizes and porosity was carried out on non-functionalized scaffolds for this study and thus a statement on the morphological properties after collagen treatment is pending at this time. Cooper et al. investigated braided biphasic ligament scaffolds made of poly(lactic-co-glycolic acid) (PLGA) microfibers and reported an optimal porosity above 50% and pore diameters of 175–233 µm, which are needed for in vivo tissue ingrowth [62]. Pore sizes between 200–250 µm and a porosity greater than 80% have been reported for bone tissue engineering [63,64]. Spalazzi et al. developed a knitted scaffold (consisting of polyglactin, PLGA and bioactive glass) with a triphasic structure implemented with an additional sintering process that exhibited porosities between 25–60% for bone, interface and ligament phase, respectively [65]. Incorporating an interface-specific region directly to the scaffold design demonstrated for the embroidered structures in this study was not realized for other fiber-based scaffolds by means of braiding or knitting [66]. However, the colonization of a triphasic scaffold with different cell types or the zonal-dependent differentiation of one cell type is very complex and challenging because, among other things, the different cell culture conditions (such as medium composition, chemical and physical stimulation, etc.) must be addressed [67]. In the present study, the scaffolds were cultivated in ligamentocyte growth medium (see detailed description in the Materials and Methods section) without osteo- or ligamento-inductive components. This was very similar to the reports of other authors, such as Cooper et al., 2014, but they have also added beta-glycerophosphate to fibroblasts and osteoblasts (cell lines) co-cultured on PLA scaffolds and strived also to optimize the medium conditions to achieve a long-lasting cell vitality and ECM synthesis [68]. This can be very time-consuming. In another study, the co-culture model was achieved by using a cell suspension of osteosarcoma cells and fibroblasts on electrospun polymer gradient scaffolds with aligned and randomly orientated poly-ε-caprolactone nanofibers [54]. The disadvantages of this method include unprecise and uncontrolled cell positioning and therefore, difficulties with the observation of cell–cell interactions. A co-culture might be more stable under 3D conditions (such as spheroids or pellets) due to the fact that cells under 2D cultivation conditions showed a higher dedifferentiation rate and a quicker loss of ECM synthesis in comparison to the 3D model system [69,70]. Notably, 3D-printed enthesis scaffolds containing cells and hydrogel offer a solution to this problem as they are able to accommodate cellular differences and interactions in the tissue. However, the compressive modulus of 3D-printed constructs slowly decreased during 28 days of cultivation [71]. Alternatively, 3D-printed bone–ligament–bone scaffolds made out of polycaprolactone fibers could also be colonized with cell sheets to achieve a sufficient compartmentalization, but Lui et al., 2019 have also mentioned fatigue of the scaffolds as a limitation [72]. Therefore, in this study, an attempt was made to arrive at a satisfactory colonization strategy for biomechanically stable triphasic scaffolds as quickly and efficiently as possible. The spheroid method offered the best prerequisite to place several different cell types on one carrier to mimic the native tissue much quicker than the monolayer cultivation does [73,74]. However, so far, it could only be shown that directly produced ligament-derived cell spheroids can be used for successful colonization [59,75], but not whether this is also possible with longer-cultured lACL or osteogenically differentiated hMSC spheroids. At this point, it should be mentioned that this is the first approach to show the feasibility of co-culturing osteogenically induced and ligamentogenic cell types on a novel multiphasic cell carrier, and therefore multifaceted analysis, including qualitative vitality assay, immunocytochemistry, calculation of DNA and GAG contents, real-time PCR and classical histological staining were performed but certainly not all possible methods (such as transcriptome analysis, single-cell RNA sequencing or fluorescence in situ hybridization [76]) have been applied to investigate the full breadth and depth of the matter. Vitality assay was only performed in a qualitative way, similar to other authors [32,68,71], to acquire not only a direct answer regarding cell vitality after culturing, but also to visualize and localize exactly which cells are still alive. Hence, so far no qualitative vitality analysis, such as the Presto Blue Assay [33], the ((3-(4,5-dimethylthiazol-2-yl)-2,5-diphenyltetrazolium bromide) (MTT) Assay [77] or the fluorescent Alamar Blue Assay, was included, which will be performed in future. Bone-marrow-derived MSCs were chosen for this study due to their easy access and their well-established higher osteogenic and chondrogenic differentiation capacities in contrast to adipose-tissue-derived stem cells [78]. Finally, it could be shown that by means of spheroids, a targeted colonization of individual zones is not only possible, but also an adherence, outgrowth and spreading of the cells on the filaments/threads could be demonstrated, and these spreading cells could also colonize the material surface evenly via emigration from the spheroids. Based on the collagen type I gene expression and the expression of sGAGs, it could be shown, in agreement with other studies, that a 14-day long-term culture is recommendable [79]. Especially for the hMSC spheroids, 14 days of osteogenic pre-differentiation seemed to be essential for ECM production. The calcium deposition of osteogenically differentiated hMSC spheroids on the bone zone of the triphasic scaffold could be observed. To target even more rapid and successful osteogenic differentiation, additional stimulation with a bone morphogenic protein (BMP-2) may be considered in the future [80]. Small nucleolar RNA host gene (SNHG)14 is a newly discovered long non-coding RNA that could induce osteogenic differentiation of hMSCs in vitro by targeting miR-2861 [81]. Although it would be much easier to use only hMSCs (one cell source) for all three zones, the ligamentous [82] and fibrocartilaginous [83,84,85] differentiation of hMSCs is very controversially discussed. In contrast to another study, where the colonization of a decellularized scaffold was performed with a zonal application of growth factors [53], no supplements except for ascorbic acid in the growth medium to stimulate collagen synthesis were used in this study. Therefore, additives were omitted in this study to demonstrate a possible effect of the embroidery pattern (porosity and pore size) and functionalization with collagen foam as well as gas-phase fluorination on the differentiation but also on the maintenance of the cellular phenotype. Other studies have also shown that porosity has a significant influence on the differentiation of stem cells [86,87]. Although the filaments were partially covered by a collagen foam, it could be seen not only in the immunocytochemical staining but also in the histological analysis that the migrating cells orientate themselves to the direction of the filaments in order to build up an ordered ECM structure. Especially (as also used in this study) static cultivation seemed to be not very useful for successful ligament/tendon development [88]. However, the initial adhesion of the spheroids on the scaffold needs a static phase of culture. Until now, there has been no clear consensus concerning the cultivation of ligamentous spheroids, and therefore there has been no clear consensus concerning their ECM synthesis and maturation for ligament tissue engineering [89,90]. For this reason, we chose the same pre-cultivation (14 days) as for the hMSC spheroids, and we saw that the long-time pre-cultivation of lACL spheroids was possible, supported by the maintained cell vitality, spheroid roundness and sGAG production. The sGAG production of the hMSC spheroids on our scaffolds was also upregulated with increasing time, indicating that an ECM formation was in progress. The sGAGs, especially biglycan and chondroitin sulphate, belong to the major components of the bone and are of central interest to achieve a tough bone construct [91]. The sGAGs, such as decorin, biglycan and aggrecan, also play a decisive role in the physiology and biomechanics of tendons and ligaments [92]. The sGAG increase in the ligament zone in long-term culture that we have demonstrated is of particular importance for successful functional ACL reconstruction in the future. In contrast to the sGAG amount, but also in contrast to other studies [93], which showed an increase in the DNA amount with increasing cultivation time in the different co-cultured zones, we observed not only in the bone but also in the ligament zone a decrease in DNA. The reduced DNA content in both the lACL and the hMSC spheroids from the long-term cultures is possibly influenced by the rather large spheroid size [94] and thus, the reduced oxygen supply to the cells. The directly formed spheroids of both cell types possessed a mean diameter larger than 500 µm. Oxygen is a key factor that directly affects stem cell proliferation and differentiation in vitro and in vivo [95]. The downregulated DNA amount and therefore, the reduced cell proliferation in long-time cultivated hMSC spheroids could also be due to the reduced relative expression of RUNX2 [96]. Differences in proliferation have also been visualized through immunocytochemical staining of the proliferation marker Ki-67. The transcription factor RUNX2 belongs to a particular DNA binding and protein–protein interaction domain called runt box (abbreviated RUNX2). RUNX2 is required for osteogenesis [96,97]. The upregulation of the expression of bone ECM genes, such as collagen type I, osteopontin, bone sialoprotein, osteocalcin and fibronectin, is induced by RUNX2 due to the activation of their promotors [96]. The osteogenically differentiated spheroids in long-term culture were still able to express the important bone ECM components such as collagen type I and the hypertrophic marker collagen type X. A detailed observation of further markers (such as alkaline phosphatase [98], bone sialoprotein [33,93], osteocalcin [98], osteonectin [32], osteopontin [99], collagen type V [100,101] and the bone-specific transcription factor bone-morphogenic protein 2 [102]) will be performed in future. Since these two ECM components could also be detected in non-differentiated hMSC spheroids (directly colonized), it can be assumed that the scaffolds also have an osteoconductive influence due to their collagen foam functionalization [103]. The progress of ossification was demonstrated through histological Alizarin Red staining and can be quantified through further staining similar to Von Kossa staining and quantitatively measured with a calcium content Assay [104]. The relative gene expression of the most prominent ECM components collagen type I and tenascin C could also be detected, which are important for ligament regeneration [105,106]. Furthermore, MKX was investigated due to the fact that this tendon/ligament-related transcription factor is very important for ECM maturation in early development and also in the adult stage for tendon/ligament homeostasis and is a regulator for collagen type I formation [107]. In the ligament structure, the investigation of collagen type III, tenomodulin, decorin and scleraxis will be added to prove maintenance of ligamentogenic differentiation. However, it has already been demonstrated that lACL fibroblasts as a monolayer but also as self-assembled spheroids were able to express these ligament-specific markers on the ligament zone [44,59]. A distinction between the newly formed collagen fibers and the collagen foam from the functionalization could not be obtained through immunocytochemical staining, which does not allow one to distinguish between both. The functionalization with gas-phase fluorination and collagen foam had no serious effects on the mechanical properties of the scaffold in vitro [59] or in vivo [42]. Tissue-specific scaffold functionalization using ligament and cartilage ECM components restricted to specific scaffold zones has been reported previously by Olvera et al. (2020) as a method to achieve a multiphasic construct. The authors showed a zone-specific chondro-/ligamentogenic differentiation of MSCs on PCL microfibers due to zone-dependent functionalization of the fibers [34]. A direct comparison to the native ACL was not performed, due to the small size of the native rabbit ACL structure and the macroscopically not-clearly-separable enthesis zones [49]. Nevertheless, in the triphasic scaffolds used in the present study, only the topology (fiber orientation and pore sizes) differed in the structural zones, but the functionalization with collagen foam and fluorination was uniform for the entire scaffold. The idea was not to apply additional influence factors to obtain a first impression of cell response to the scaffold structure.

One of the biggest challenges for enthesis reconstruction is the restoration of an ordered and directed ECM formation in order to be mechanically resilient again. For this reason, embroidered scaffolds with mainly longitudinally orientated polymer fibers were used to allow the cells and thus also the newly formed collagen fibers to align to the main direction of future tension in the native ACL tissue to be reconstructed. Therefore, the co-colonized triphasic scaffolds should be mechanically stretched in future to observe the influence on cells with their functional ECM syntheses but also the degradation of the polymer fibers. This should be performed not only in vitro under biophysical conditions but also in vivo in the rabbit model to obtain a closer look at the material–tissue integration, the cell–cell interaction during movement, the influence of immune cells and the body fluid and the immigration of surrounding tissue cells. In vivo implantation is of utmost importance for a better cellular understanding and the next step towards clinical trial and will be performed in future. 

## 4. Materials and Methods

### 4.1. Preparation of Embroidered P(LA-CL)/PLA Triphasic Scaffolds

Two different thread materials were used for scaffold fabrication: a monofilament suture made of P(LA-CL) (USP 7-0, Gunze Ldt., Osaka, Japan) and a melt-spun PLA multifilament consisting of six single filaments (Tt = 155 dtex, Ingeo biopolymer 6202D, NatureWorks, Minnetonka, MN, USA, fiber melt spinning at the Leibniz-Institut für Polymerforschung Dresden e. V. (IPF, Dresden, Germany). A water-soluble non-woven made of polyvinyl alcohol (PVA, Freudenberg Einlagestoffe KG, Weinheim, Germany) was used as an embroidery base material on the embroidery machine (Type TLMX-901, Tajima Industries, Nagoya, Japan). Later, the base material was washed out by rinsing three times for 30 min in pyrogen-reduced water on a compact shaker (KS 15 A, Edmund Bühler GmbH, Bodelshausen, Germany). After that, the remaining porous scaffolds were dried at room temperature (RT). The triphasic scaffold design was realized by overlaying the bony (green) and ligamental (blue) pattern areas in an undulating design (as shown in the graphical abstract, representing one ply). The bone area (5 mm length) exhibited a triaxial pattern with 1.8 mm stitch length, a stitch density of 1.9 stitch/mm^2^ and P(LA-CL) serving as upper and lower threads (Figure 1A, region a). For the ligament pattern (8 mm length), a zig-zag design with 1.8 mm stitch length, 15° stitch angle and 0.2 mm duplication shift were used with P(LA-CL) as upper and PLA as lower threads (Figure 1A, region c). The enthesis part was created with the triaxial pattern for the bone area (P(LA-CL) as upper and lower threads) and a partial overlap of undulated line stitches (P(LA-CL) as upper, PLA as lower thread) (Figure 1A, region b). To create a 3D scaffold for all further examinations, three plies were stacked and locked together.

### 4.2. Determination of the Porosity Using Computed Tomography

The morphological structures of two triphasic embroidered scaffolds were studied with micro-computed tomography (µCT) before functionalization. The µCT measurement was realized with a laboratory µCT device (CT-ALPHA, ProCon X-ray GmbH, Sarstedt, Germany) using X-ray energy at 50 keV and a tube target current of 110 µA. The samples were scanned within 2300 positions in a 360-degree trajectory. For a single position, three absorption pictures were averaged to reduce the signal-to-noise ratio. With the help of the CT reconstruction software X-AID (Version 2022.12.3, MITOS GmbH, Garching, Germany), a spatial image resolution of 3.5 µm was reached. The porosity was quantified using software VG Studio MAX - (Version 3.5, Volume Graphics GmbH, Heidelberg, Germany). After thresholding the X-ray absorption values for polymer fibers, the resulting structure were analyzed within a three-dimensional region (900 × 900 × 300 pixel) using the VG Studio add-on foam/powder analysis. For a better comparison of the discovered irregular pore volumes, an equivalent sphere diameter was used. The statistical analysis of the values was performed using Origin (Version 2021, OriginLab Corporation, Northampton, MA, USA).

### 4.3. Functionalization of Embroidered Scaffolds

Gas-phase fluorination was performed at the FILK Freiberg Institute (Freiberg, Germany) in a fluorination batch reactor (Fluor-Technik-System GmbH, Lauterbach, Germany) using a mixture of 10% fluorine gas in synthetic air for 60 s. After the fluorination process, the scaffolds selected for additional collagen functionalization were flushed with synthetic air prior to immersion in solved bovine acid collagen and subsequently refibrillated with phosphate buffer and NaCl. The resulting collagen hydrogel pervading the embroidered scaffold was desalted and lyophilized to form a porous foam between the threads. The collagen cross-linking was executed with HMDI in a gas-phase in a desiccator.

### 4.4. hMSC Isolation

The hMSCs were isolated from 8 voluntary bone marrow donors (five females and three males with an average age of 24 ± 2.2). The use of human tissue samples was approved by the ethical commission of the Bavarian medical association (no. 17074, 6 February 2018) and was in accordance with the declaration of Helsinki. From the iliac crest, bone marrow blood (Nuremberg General Hospital) was obtained and transferred with MSC growth medium (1:4; Dulbecco’s modified Eagle’s medium (DMEM with stable glutamine [3.7 g/L NaHCO_3_ and 4.5 g/L D-glucose] (PAN-Biotech GmbH, Aidenbach, Germany)), 5% human-growth-factor-rich human Platelet Lysate (PL) solution (PL BioScience, Aachen, Germany), 0.04% heparin (PL BioScience), 1% [*v*/*v*] amphotericin B, 1% natrium pyruvate (PAN-Biotech) and 1% penicillin/streptomycin (PAN-Biotech) into plastic cultivation flasks (T175, Sarstedt AG & Co. KG, Nümbrecht, Germany). A total of 10 mL of fresh medium was added after three days, and after an additional three days, the adherent cells were carefully washed twice with phosphate-buffered saline (PBS, PAN-Biotech) and then incubated with 20 mL fresh medium until a confluency of 80–90% was reached before they were used to form spheroids or for further expansion culture.

### 4.5. lACL Isolation

The lACL-derived ligamentocytes were harvested from five adult males, healthy 12-month-old ACLs of New Zealand Rabbits, obtained from the regional slaughterhouse. The explanted LACLs were split into 2 mm^2^ pieces and were cultivated in T25 culture flasks in ligamentocyte growth medium (Dulbecco’s Modified Eagle’s Medium [DMEM]/Ham’s F12 medium [1:1, PAN-Biotech] supplemented with 10% fetal bovine serum [FBS, PAN-Biotech], 1% penicillin/streptomycin solution [PAN-Biotech], 25 μg/mL ascorbic acid [Sigma-Aldrich, Munich, Germany], 2.5 μg/mL amphotericin B [PAN-Biotech] and MEM amino acid solution [Sigma-Aldrich]). After 7–10 days, ligamentocytes emigrated and were expanded using 0.05% trypsin/0.02% ethylenediaminetetraacetic acid (EDTA, PAN-Biotech) for the subsequent experiments.

### 4.6. Spheroid Preparation and Long-Time Pre-Cultivation

Each spheroid contained 5.0 × 10^4^ cells and was fabricated by pipetting 400 µL cell suspension in 0.5 mL tubes and centrifuged (5.000× *g* for 5 min) before the tubes were incubated at 37 °C for two days. After two days of formation, the spheroids detached from the tube wall and were at the bottom where they could be harvested for scaffold colonization or further cultivation. For long-time pre-cultivation (LC), lACL spheroids as well as hMSC spheroids were transferred into non-adherent 96-well plates (flat bottom TC-plate 96 well Cell + F, Sarstedt). LACL spheroids were further cultivated for 14 days in ligamentocyte growth medium. hMSCs were cultivated in 100 µL/well osteogenic differentiation medium (consisting of DMEM, Dexamethason [VWR, Darmstadt, Germany], Glyerol-3-posphate [VWR], L-ascorbic acid [Sigma-Aldrich], 1 molar HEPES [Merck KGaA, Darmstadt, Germany], 5% PL solution and 1% penicillin/streptomycin) for a further 14 days. Medium change was performed every second day during the whole cultivation period. Macroscopical images were taken to evaluate the spheroid shape and size during cultivation.

### 4.7. Triphasic Scaffold Colonization 

Triphasic scaffolds were disinfected for a minimum of 30 min in 70% ethanol (EtOH, AppliChem GmbH, Darmstadt, Germany), washed three times in pyrogen-free, sterile, hypotonic dist. water (Carl Roth) and then pre-incubated in FBS for 30 min. Sterilized scaffolds were cultivated with lACL and hMSC spheroids directly after formation (called “direct culture” (DC)) or with spheroids pre-cultivated for 14 days (LC). Bone (31,250 ± 1822 cells per mm^2^) and ligament (32,712 ± 2203 cells per mm^2^) scaffold zones were colonized. The colonized scaffolds were further statically cultivated for 14 days at 37 °C and 5% CO_2_ in lapine ligamentocyte growth medium before they were analyzed.

### 4.8. Cell Survival 

The vitality of the spheroids was checked before scaffold colonization. The colonized triphasic scaffolds were longitudinally halved, incubated in 50 µL vitality staining solution (5 µL/mL fluorescein diacetate (FDA, Sigma-Aldrich, 3 mg/mL dissolved in acetone as a stock solution) and 1 µL/mL propidium iodide (PI, 1% solution, Carl Roth GmbH and Ko.KG) in phosphate buffered saline (PBS, PAN-Biotech)) for one minute at room temperature (RT). Viable cells were green and dead cells were red, as visualized using Confocal Laser Scanning Microscopy (CLSM, SPE-II, Leica Microsystems GmbH, Wetzlar, Germany). The spheroid vitality was calculated based on the vital (green channel) and the dead (red channel) with the LeicaX 3D software (3.5.7.23225). Based on the pictures showing only vital cells, the colonized area of the triphasic scaffold total area was measured with the LeicaX 3D software. Five independent experiments were performed. 

### 4.9. Immunocytochemistry 

Protein expression of collagen type I, vimentin, F-Actin and Ki-67 was assessed by using CLSM. Co-cultured scaffolds were fixed in 4% paraformaldehyde (PFA, Morphisto GmbH, Offenbach am Main, Germany) for 30 min and washed with Tris-buffered saline (TBS: 0.05 Tris, 140 mM NaCl, pH 7.6, Carl Roth GmbH, Karlsruhe, Germany) before incubation with blocking buffer (5% protease free donkey serum diluted in TBS with 0.1% Triton X 100 for cell permeabilization) was performed for 20 min at RT. DC and LC triphasic scaffolds (for each experiment longitudinally cut) were incubated with a primary antibody (see Table 1) overnight at 4 °C. The samples were rinsed with TBS and then incubated for 1 h with a secondary antibody (Table 1) at RT. The cell nuclei were counterstained using 10 μg/mL 4′,6′-diamidino-2-phenylindol (DAPI, Roche, Mannheim, Germany). Phalloidin-Alexa-Fluor 488 (1:100, Santa Cruz Biotechnologies, Inc., Dallas, TX, USA) was used to depict the filamentous (F)-actin cytoskeletal architecture. Samples were examined using CLSM after three washing steps with PBS.

### 4.10. Quantitative Measurement of Total DNA and sGAG Content 

The DNA contents of the scaffolds were measured using CyQuant assay (Invitrogen, Waltham, MA, USA) according to the user manual including calf thymus DNA as a standard. Each scaffold was separated with a scalpel into the bone and the ligament zone (the transition zone was cut out due to the fact that it was not selectively colonized and then homogenized with a 7 mm stainless steel bead (RNase and DNase free, sterile, Qiagen, Hilden, Germany) by using TissueLyser LT (Qiagen, Hilden, Germany, 50 Hz, 5 min, RT). The samples were digested using a proteinase K solution (0.5 mg/mL, Carl Roth GmbH and Ko.KG) and dissolved in 50 mM Tris/HCl (1 mM EDTA, 0.5% Tween20, pH 8.5) for 16 h at 56 °C under continuous shaking (36 rpm) before centrifuged for 30 min at 10,000 rpm. The supernatants were frozen at −20 °C for 30 min to stop the enzymatic reaction. A 10 µL sample and 150 µL TE buffer were mixed. Triplicates of specimens (each 25 µL) were transferred to a black 96-well plate with a flat bottom (Brand GmbH, Wertheim, Germany), and 25 µL dye solution (1× HBSS + dye solution 1:250) was added. Plates were protected from light and incubated at 37 °C for 60 min. The fluorescence was measured at λ = 485 excitation/λ = 530 emission using a fluorometric plate reader (Infinite M200, Tecan Austria GmbH, Grödig, Austria).

The lysates were diluted in phosphate-buffered EDTA (PBE) buffer (100 mM Na_2_HPO_4_ and 5 mM EDTA, pH 8.0) to quantify the sulphated glycosaminoglycan (sGAG) contents before the dimethylmethylene blue (DMMB, AppliChem, Darmstadt, Germany) dyeing solution (8.9 mM DMMB hydrochloride in 600 mg glycerine, 467 mg NaCl and 200 mL dist. water) was added. The absorption shifts from λ = 525 nm to λ = 595 nm were measured. Total sGAG content was calculated based on chondroitin sulphate sodium salt from shark cartilage (Sigma-Aldrich) as a standard (Infinite M200, Tecan Austria GmbH).

### 4.11. RNA Isolation 

Colonized triphasic scaffolds were separated into the bone and the ligament zone (transition zone was cut out) and snap-frozen (each *n* = 5) in liquid nitrogen. The samples were homogenized with a tissue lyser (Qiagen, Hilden, Germany) at 50 Hz for 5 min. A total of 1000 µL pre-cooled Qiazol (Qiagen) was added, vortexed and incubated for 5 min at RT before 200 µL chloroform was added. After 5 min of incubation at RT, the samples were centrifuged for 15 min at 4 °C with 12,000× *g*. The clear supernatant was removed and pipetted on a Qiashredder (Qiagen). RNA was isolated and purified using an RNeasy Mini kit according to the manufacturer’s instructions (Qiagen), including on-column DNase treatment. The purity (calculated from 260/280 absorbance ratio) and quantity of the RNA samples were monitored using a Nanodrop ND-1000 spectrophotometer (Peqlab, Biotechnologie GmbH, Erlangen, Germany).

### 4.12. Relative Gene Expression

The cDNA synthesis was performed with 63 ng of total RNA and the reverse transcription kit (QuantiTect Reverse Transcription Kit, Qiagen, Hilden, Germany) according to the supplier manual. For each quantitative real-time PCR (qRT-PCR) reaction, a total of 15 ng of cDNA was used for TaqMan Gene Expression Assay (Life Technologie, Darmstadt, Germany) with primer pairs for *collagen type I* (*COL1A1*), *tenascin C* (*TNC*), *Mohawk* (*MKX*), *collagen type X* (*COL10A1*) and *runt-related protein (RUNX) 2* and the reference gene *beta actin (BAC*) for hMSCs and *glyceraldehyde 3-phosphate dehydrogenase (GAPDH)* as a reference gene for lACLs (Table 2). The real-time PCR detector StepOnePlus (Applied Bioscience (ABI), Foster City, CA, USA) thermocycler with the program StepOnePlus software v2.3 (ABI, Foster City, CA, USA) was used for qRT-PCR. The relative expression of the gene of interest by the cells on the scaffolds was normalized to the *BAC* or *GAPDH* expression and calculated for each sample using the ΔCT method as described [108]. The values of relative gene expression of LC were related to the DC, which were set to 1. 

### 4.13. Alizarin Red Staining

Alizarin Red staining was used to indicate the progress of osteogenesis due to the fact that calcium deposits could be detected. The triphasic scaffolds were rinsed 2 × 5 min with PBS at RT before undergoing fixation for 60 min in 60% isopropanol. After rinsing in TBS, the scaffolds were stained with Alizarin Red solution (Sigma Aldrich) for 5 min. The scaffolds were dehydrated in acetone before they were photographed with a light microscope.

### 4.14. Statistical Analyses

All calculated values from the spheroid vitality, the colonized area, the DNA content, sGAG content and relative gene expression were expressed as mean with the standard deviation (SD) from five independent experiments. Measurements of the spheroid diameter, the spheroid roundness and calcium deposits (red intensity) were performed with ImageJ (1.52 d, National Institute of Health, Bethesda, MD, USA). The statistics tests were performed, and graphics were created with GraphPad Prism 8.4.3 (686) (GraphPad Software Inc., San Diego, CA, USA). After the ROUT outlier test (1%), the normal distribution of the samples was tested with the one-way ANOVA, and multiple comparison testing (Tukey post hoc test) was performed to evaluate the significant differences between the groups. Statistical significance was set at a *p*-value ≤ 0.05 (*), *p*-value ≤ 0.01 (**), *p*-value ≤ 0.0005 (***) and *p*-value ≤ 0.0001 (****).

## 5. Conclusions

Vitalization of all zones in tissue-engineered enthesis constructs is very important for successful ACL reconstruction in the future. A spheroid-based colonization strategy allows for a precise positioning of the osteogenically differentiated hMSC and ligamentocyte spheroids on their respective enthesis zones. According to our hypothesis, spheroids of two different cell types from two different species could adhere, survive and colonize a triphasic scaffold not only after the spheroid assembly process but also after long-time pre-cultivation and osteogenic differentiation of the spheroids. The newly obtained data from this proof-of-construct study show that co-cultivation of stem cells and ligamentocytes on a scaffold is possible under the same growth medium conditions, and they provide sufficient compartmentalization. Nevertheless, the osteogenic pre-differentiation of the hMSCs did only stimulate collagen type I transcription but did not increase the gene expression of the early osteogenic marker RUNX2 and hypertrophy-associated collagen type X, hence, failing to confirm our initial hypothesis. However, the colonization strategy should be further improved in the future to achieve rapid adherence and more effective output in terms of ECM production so that bone integration in vivo can indeed be achieved. 

## Figures and Tables

**Figure 1 ijms-24-06714-f001:**
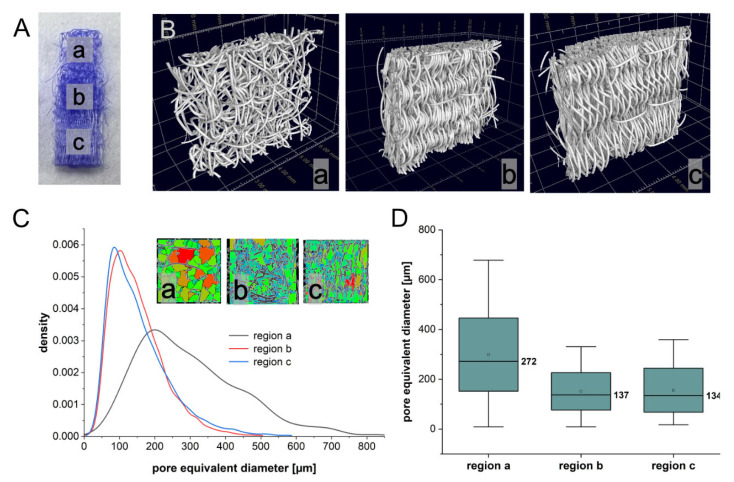
Analysis of scaffolds with micro-computed tomography. (**A**) µCT measurement regions within the triphasic scaffold. (**B**) 3D visualization of reconstructed CT images for the three regions: (**a**) bony, (**b**) transition, (**c**) ligament. (**C**) Distribution of the pore equivalent diameter representing irregular pore volumes. (**D**) Pore equivalent diameter for the different regions of the triphasic scaffold with median values.

**Figure 2 ijms-24-06714-f002:**
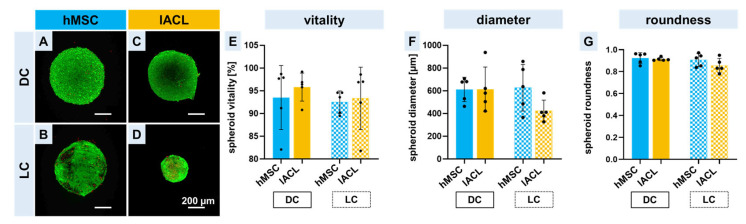
Vitality assay and shapes of spheroids prepared for scaffold seeding. Viable (green) and dead (red) human mesenchymal stem cells (hMSCs; **A**,**B**) or lapine anterior cruciate ligamentocytes (lACLs; **C**,**D**) of the spheroids two days after assembly before being seeded on the scaffolds. Direct cultivation (DC, directly after 2 days of spheroid formation) (**A**,**C**) and long pre-cultivation (LC, 2 days of spheroid formation plus 14 days of cultivation) (**B**,**D**). Relative vitality (**E**), the diameter (**F**) and the roundness (**G**) of the spheroids before putting them on the scaffolds. Five independent experiments analyzed with ROUT (1%) outlier test and one-way ANOVA showed no significant differences. Scale bars of 200 µm.

**Figure 3 ijms-24-06714-f003:**
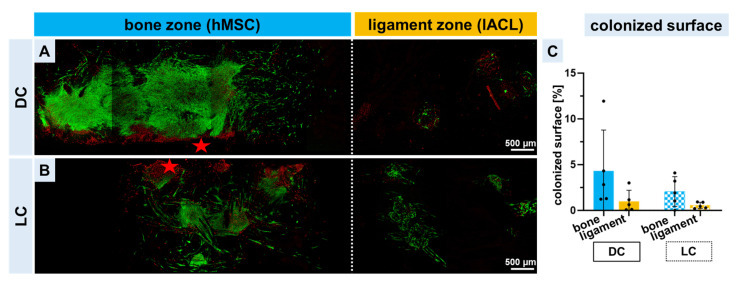
Vitality assay of spheroid-colonized scaffolds after 14 days. Viable and dead human mesenchymal stem cell (hMSC) and lapine anterior cruciate ligamentocyte (lACL) spheroids, which were directly placed on the scaffold zones immediately after the 2 days of spheroid assembly (**A**, DC) or with spheroids pre-cultured for 14 days (**B**, long-time: LC) before being seeded on the scaffold bone or ligament zones after 14 days of scaffold cultivation. Scale bars of 500 µm. The calculation of the colonized surface was based on the viability pictures (**C**). Five independent experiments analyzed with ROUT (1%) outlier test and one-way ANOVA showed no significant differences. The red stars show the lower third of the scaffold where the transsection was performed.

**Figure 4 ijms-24-06714-f004:**
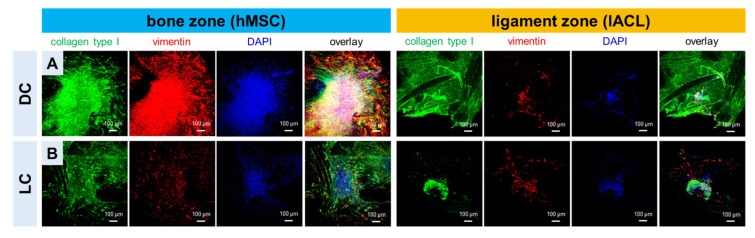
Extracellular matrix formation and vimentin expression after 14 days of scaffold culture. Immunocytochemical staining of collagen type I, mesenchymal marker vimentin and counterstaining of cell nuclei (4’, 6-Diamidino-2-Phenylindole, DAPI) of human mesenchymal stem cell (hMSC) and lapine anterior cruciate ligamentocyte (lACL) spheroids, which were directly placed on the scaffold zones immediately after the 2 days of spheroid assembly (**A**, DC) or with spheroids pre-cultured for 14 days (**B**, long-time: LC) before being seeded on the scaffold bone or ligament zones, shown after 14 days of scaffold cultivation. Scale bars of 100 µm.

**Figure 5 ijms-24-06714-f005:**
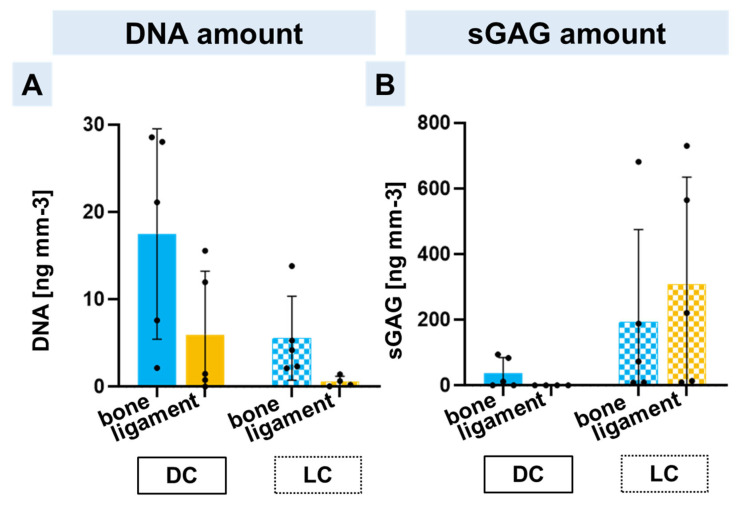
Calculation of the DNA (**A**) and sGAG (**B**) amount of each scaffold zone after direct (DC) or long pre-cultivation (LC). The DNA and sGAG amounts were calculated per cubic millimeter scaffold in the bone zone with hMSC spheroids (blue) and in the ligament zone with lACL spheroids (orange). Five independent experiments analyzed with ROUT (1%) outlier test and one-way ANOVA showed no significant differences.

**Figure 6 ijms-24-06714-f006:**
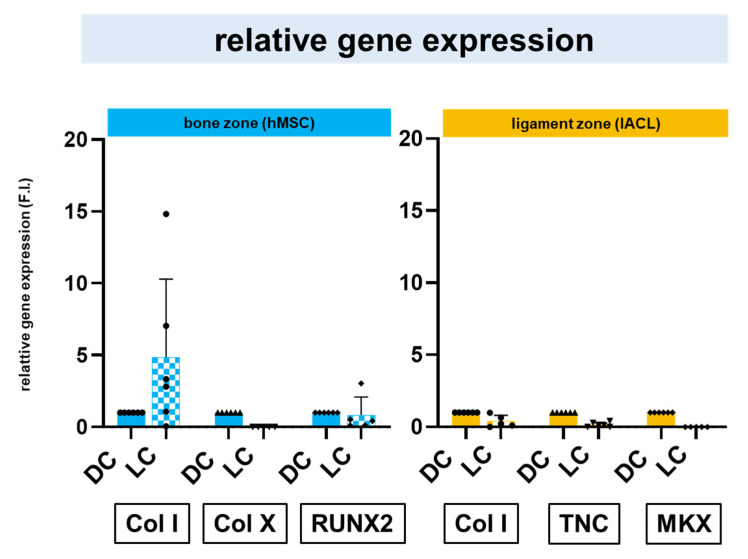
Relative gene expression of bone- and ligament-related extracellular matrix components and transcription factors. Normalized (F.I. = fold induction) gene expression of *collagen type I (Col I), collagen type X (Col X)* and the transcription factor *runt-related protein (RUNX)2* was evaluated for the directly seeded (DC, set to 1) or longer (LC) pre-cultivated (osteogenically differentiated) hMSC spheroids on the bone zone, while that of *Col I*, *tenascin C (TNC)* and the transcription factor *Mohawk (MKX)* was analyzed for the directly seeded or the longer pre-cultivated lACL spheroids. Bone and ligament zones were separated before evaluation. Five independent experiments analyzed with ROUT (1%) outlier test and one-way ANOVA showed no significant differences.

**Figure 7 ijms-24-06714-f007:**
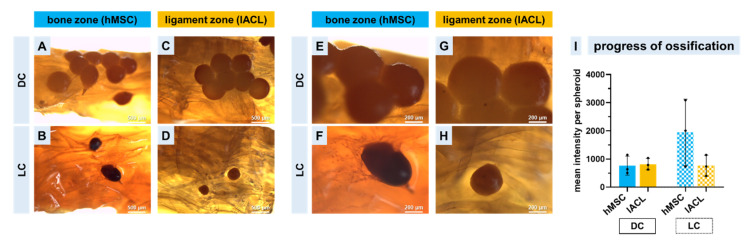
Calcium deposition in spheroids co-cultured on the bone and ligament zones of triphasic scaffolds after a cultivation time of 14 days. The scaffolds were colonized with spheroids directly after assembly (DC) or with long-time pre-cultivated (LC) spheroids. The Alizarin Red stain is shown in an overview (**A**–**D**) and at higher magnification (**E**–**H**). The progress of calcium deposition in the spheroids within the stained scaffolds was evaluated by measurement of the mean red intensity per spheroid (**I**).

**Table 1 ijms-24-06714-t001:** Antibodies used for immunocytochemistry.

Target	Primary Antibody	Dilution	Secondary Antibody	Dilution
Collagen type I	Goat anti-human, Biozol	1:30	Alexa Fluor 488, Donkey-anti-goat, Invitrogen AG	1:200
Vimentin	Monoclonal mouse-anti-vimentin, Dako	1:40	Indocarbocyanine Cy3, Donkey-anti-mouse, Invitrogen AG	1:200
F-Actin	Phalloidin-Alexa-Fluor 488, Abcam	1:100		
Ki-67	Mouse-anti-human, Merck	1:30	Donkey-anti-mouse Cy3, Invitrogen	1:200

Cy3: cyanine-3.

**Table 2 ijms-24-06714-t002:** Primer list for gene expression analysis.

Gene Symbol	Species	Gene Name	Amplicon Length	Assay ID	Efficacy
*COL1A1*	*Homo sapiens*	Collagen type I alpha 1	66	Hs00164004_m1	2.06
*COL10A1*	*Homo sapiens*	Collagen type X alpha 1	76	Hs00166657_m1	2.15
*RUNX2*	*Homo sapiens*	Runt-related protein	116	Hs00231692_m1	1.94
*BAC*	*Homo sapiens*	Beta actin	171	Hs99999903_m1	1.89
*COL1A1*	*Oryctolagus cuniculus*	Collagen type I alpha 1	70	Oc03396073_g1	1.94
*TNC*	*Oryctolagus cuniculus*	Tenascin C	61	Oc06726696_m1	1.83
*MKX*	*Oryctolagus cuniculus*	Mohawk homeobox	60	Oc06754037_m1	1.83
*GAPDH*	*Oryctolagus cuniculus*	glyceraldehyde 3-phosphate dehydrogenase	82	Oc03823402_g1	1.95

All primers were obtained from Applied Biosystems.

## Data Availability

Not applicable.

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
