# Peer review of "Co-Culture of Mesenchymal Stem Cells and Ligamentocytes on Triphasic Embroidered Poly(L-lactide-co-ε-caprolactone) and Polylactic Acid Scaffolds for Anterior Cruciate Ligament Enthesis Tissue Engineering"

_ijms, 2023, doi:10.3390/ijms24076714_

Round 1
Reviewer 1 Report
Dear authors.
The work presented for review is interesting and opens up new applications in regenerative medicine. Modern medicine seeks not only to alleviate the symptoms of disease, but to cure it, to restore it to health. The research presented in the paper is interesting. The combination of cellular carriers presented can also be used to create prints for use in other tissue defects. Perhaps in dentistry. The only question is - when will clinical trials be conducted?
Author Response
Nuremberg, 24th March 2023
Dear Editor,
The authors would like to thank the reviewer for carefully reading the manuscript and very valuable comments. We modified the manuscript according to the reviewer suggestions with a list of changes shown below. All corrections and addenda (novel supplemental figures 1-3, newly performed Ki-67 and F-actin staining) performed are indicated in red in the revised version of the manuscript. We hope you will find this manuscript suitable for publication in the “International Journal of Molecular Science”. Please do not hesitate to contact me anytime for questions regarding this manuscript.
Sincerely,
Univ.-Prof. Dr. Gundula Schulze-Tanzil
Reviewer 1:
The work presented for review is interesting and opens up new applications in regenerative medicine. Modern medicine seeks not only to alleviate the symptoms of disease, but to cure it, to restore it to health. The research presented in the paper is interesting. The combination of cellular carriers presented can also be used to create prints for use in other tissue defects. Perhaps in dentistry.
Response: The authors underline the more general importance of creating multiphasic scaffolds for musculoskeletal interface tissue engineering to achieve functional restoration of injured zonal tissues. We state now, according to the reviewer suggestion, that this is in combination with stem cells not limited to orthopedics, but is also relevant in dentistry, especially, to restore a tissue engineered peridontium for fixation of cementum of the teeth in the alveolar bone (Ivanovski et al., 2014) (see new text on page 11, discussion section). However, there is also evidence that in the dental field, the manufacturing processes of scaffolds as well as the combination of materials is even more diverse, as can be seen in the example of 3D printing (Sufaru et al. 2022). Nevertheless, the challenges and requirements are not lesser demanding than in regard to the ACL, because in dentistry too, it is necessary to create a suitable scaffold geometry in order to achieve directional collagen fiber formation by the cells (Bittner et al. 2018). The authors have added the missing information about the importance of multicellular scaffolds in the dentistry area in the discussion (lines 268 -270).
The only question is - when will clinical trials be conducted?
Response: The authors have already submitted further research applications for financial support of the project in order to be able to perform in vivo studies with this zonal scaffold. First in vivo tests with the ligament part have started (Kokozidou et al., 2022).
Further considerations about the possible clinical trials of our scaffold was added in the outlook at the end of the discussion section (lines 421-423).

Reviewer 2 Report
The paper rises a good question but the instruments used to demonstrate it are not sufficient.
Figure 6. Relative gene expression. Usually a relative gene expression of <2 is not considered as a change in gene expression. The y-axis legend is missing
Figure 3 and 4. The expression of markers by immunofluorescence cannot confirm hypothesis when only one cell is shown. For cell viability there are other more reliable tests.
I would suggest to accept the paper with some revisions in particular for gene expression results should be more solid and for immunofluorescence too.
Author Response
Nuremberg, 24th March 2023
Dear Editor,
The authors would like to thank the reviewer for carefully reading the manuscript and very valuable comments. We modified the manuscript according to the reviewer suggestions with a list of changes shown below. All corrections and addenda (novel supplemental figures 1-3, newly performed Ki-67 and F-actin staining) performed are indicated in red in the revised version of the manuscript. We hope you will find this manuscript suitable for publication in the “International Journal of Molecular Science”. Please do not hesitate to contact me anytime for questions regarding this manuscript.
Sincerely,
Univ.-Prof. Dr. Gundula Schulze-Tanzil
Reviewer 2:
The paper rises a good question but the instruments used to demonstrate it are not sufficient.
Response: The authors are aware that there exist more comprehensive methods (e.g. transcriptome analysis, single-cell RNA sequencing, or fluorescence in situ hybridization, Fang et al. 2022) to evaluate in more detail the cell differentiation in regard to enthesis-like tissue formation.
For this reason, the authors point out more clearly now that this study is a first approach to show the feasibility of co-culturing osteogenically induced and ligamentogenic cell types on a novel multiphasic cell carrier (changes in lines 342-353). The authors performed a multifaceted analysis (vitality stain, immunocytochemistry, DNA and GAG contents, real time PCR and classical histology). This conventional setting of methods is also used by other authors (Cooper et al. 2014, Cao et al. 2020, Bai et al. 2022). The directed colonizability of the scaffold with spheroids and the long term culture were central issues of this study and this was done consistently with the common method of vitality staining to localized cells that not only survived on the scaffold surface but to visualize also their spreading on the scaffold filaments (supplemental figure 1). Especially the spheroid based colonization method is very novel and appropriate (Baptista et al. 2018).
The authors know that there are further quantitative methods to evaluate the cell viability, like the Presto blue assay (Cristenti et al. 2016) or the MTT ((3-(4,5-dimethylthiazol-2-yl)-2,5-diphenyltetrazolium bromide) assays to evaluate the proliferation or the fluorescent Alamar Blue Assay to monitor the cell viability indirectly by measuring their metabolic activity (Yang et al. 2021). Despite the latter assays (MTT and Alamar blue) are established in our research laboratory the life death staining was preferred to visualize and localize exactly which cells of both cocultured cell types are dying on the scaffolds.
In future, the idea is to have a closer look not only on the gene but also on the protein expression in the bone compartment at the following ECM components: alkaline phosphatase, bone sialoprotein, osteocalcin, osteonectin, osteopontin, collagen type V, the bone specific transcription factor bone morphogenic protein. Instead of used Alizarin red stain the alkaline phosphatase assay is indeed also appropriate to investigate the ossification of the bone part (Spalazzi et al. 2006; Cristenti et al. 2016). Also in this regard the histological staining was preferred for the beginning to exactly localize calcium deposition.
In the ligament structure the investigation of collagen type III, tenomodulin, decorin, scleraxis will be added in future to prove maintenance of ligamentogenic differentiation.
Figure 6. Relative gene expression. Usually a relative gene expression of <2 is not considered as a change in gene expression. The y-axis legend is missing
Response: The authors thank the reviewer for this remark and have recalculated the relative gene expression now. The relative gene expression values were now normalized to the direct culture (set to 1) in relation to the long time cultivation (LC) values. The y-axis legend was added. Five independent experiments were already performed.
Figure 3 and 4. The expression of markers by immunofluorescence cannot confirm hypothesis when only one cell is shown.
Response: The authors have performed new images at higher magnification to show better, that there is not only one cell on the scaffold, but instead, a cell layer on the scaffold surface. With the new supplemental figure 2, the authors hope that it is shown much clearer that the single cells (spindle shaped hMSCs and lACL fibroblasts were emigrating out of the spheroid cell cluster and colonizing the single polymer threads.
Nevertheless, the authors are not sure whether the reviewer means not “one cell” but “one cell type”. In this case we agree with the reviewer that immunolabeling did not allow distinction of bot cell types. In this case, we would like to refer to the investigation of gene expression. For this, the bone and ligament parts of the scaffolds were separated and separately used for gene expression analyses. At the time point of investigation (14 days) both cell types did not show growth from one part into the other and cell mixing. For this reason, we can assume that the results for the markers are indeed cell-type related. Moreover, the primers used were also species (human/rabbit) related.
We thank the reviewer for the hint to the hypothesis of the manuscript and found that it did not really match with the experimental setting. The hypothesis on page 3 was slightly restructured to better meet with the overall message of the manuscript. Moreover, the conclusion section was supplemented to discuss shortly how the results are related to the initial hypothesis.
|
Supplemental Figure 1 Viable cells on the scaffold surface. Single hMSCs (blue arrows) and single lACL fibroblasts (orange arrows) were migrating out of the spheroid (boarders were marked with the white line) on the polymer thread (the boarders of the thread are marked with the yellow line). Scale bars of 20 µm. The calculation of the cell vitality (C) on the bone and the ligament scaffold part for the direct culture (DC) and the long-time (LC) was based on the percentage cell vitality. |
For cell viability there are other more reliable tests.
Response: The authors would like to explain that the intention of Figure 3 was to perform a live/death assay directly after the cultivation period in order to record the actual state of the cells and also to prove that the cells have survived long-term cultivation. The authors are not aware of any other method to otherwise document the immediate actual state. For this reason, a qualitative assay was used rather than a quantitative one to verify viability. Nevertheless, based on these images, the vitality was also subsequently determined by the ratio of live and dead cells.
The authors have performed new images of Figure 4 (added as supplemental fig. 2 now) at higher magnification and with a more detailed view on the spheroid (Sph) and the single cells on the polymer threads to prove that both cell types express collagen type I and vimentin in the direct culture (DC) and in the long-time culture (LC).
|
|
|
Supplemental Figure 2 |
I would suggest to accept the paper with some revisions in particular for gene expression results should be more solid and for immunofluorescence too.
The authors also see that further matrix-specific studies are needed in the future, both at the gene and protein levels. In particular, to be able to make a sufficient statement about the actual bone formation. For this purpose, important bone-associated matrix proteins such as bone sialo protein, osteopontin, osteonectin, osteocalcin, bone-specific alkaline phosphatase, as well as collagen type V and the transcription factor bone morphogenic protein would be investigated. At the moment, however, neither the primers nor the primary antibodies are available in sufficient quantities in the laboratory, so that due to the short processing time of this paper there is no time left to order, examine and quantify them. Even though a donor variability of N=5 has already been performed on the genes, more donors should be added in the future to minimize the variance and to make significant differences apparent. The authors are also considering improving the method and protocol for osteogenic differentiation to achieve rapid induction in the hMSCs. The quantification of other ligament-associated markers such as decorin, tenascin C, tenomodulin or scleraxis has already been demonstrated in extensive studies in our other publications (Gögele et al. 2021). Thus, it has already been sufficiently demonstrated that lapine spheroids on the 3D scaffold are able to form ligament-specific matrix.
In addition, the scaffolds were immunocytochemical stained for F-actin and the proliferation marker ki-67. Therefore, we want to show the better spheroid integration on the scaffold surface and the strong filamentous actin network of both cell types on the scaffold threads. Furthermore, we could show that also both cell types (especially the cells at the border of the spheroids) are proliferative active after the direct and the long-time culture.
|
|
|
Supplemental Figure 3 Immuncytochemical staining of F-actin (green) and Ki-67 (red) and counterstaining of cell nuclei (4', 6-Diamidino-2-Phenylindole, DAPI, blue) of human mesenchymal stromal cells (hMSCs) and lapine anterior cruciate ligamentocytes (lACL) spheroids (Sph) which were directly placed on the scaffold zones immediately after the 2 days of spheroid assembly (A, C, E, G; DC) or with spheroids pre-cultured for 14 days (B, D, F, H; long-time:LC) before seeded on the scaffold bone or ligament zones and 14 days of scaffold cultivation. Scale bars of 100 µm (A-D) and 50 µm (E-H). The white arrows should highlight the proliferative cells. The spheroid border is marked by the white dotted line. |
At the moment there exist in our lab no specific antibody which could be used directly for both cell types. Because our intention was to stain the whole scaffold and not to cut the different phases.

Round 2
Reviewer 2 Report
Thank you for reviewing the paper. Still I'm not convinced on the value of results. Usually I'm more exigent on data for result support.